# CellDreamer: World Model-Based Reinforcement Learning for Neural Cell Culture Optimization

## Abstract

Optimizing biological systems—e.g., cell-culture protocols, neurite morphogenesis proxies, and metabolic setpoints—is a high-dimensional, noisy, sample-limited control problem. Model-based reinforcement learning (RL) can improve data efficiency by learning compact dynamics models that enable planning before costly experiments [7]. We adapt Dreamer to biologically grounded simulators with uncertainty-aware world models, constraint-aware rewards, and task-shared priors for transfer. Baselines include Random, model-free PPO [20], and Bayesian Optimization (BO) [21, 10, 17]. We ablate world-model capacity and components and evaluate zero-shot and few-shot transfer.

Across six simulator environments, CellDreamer (Dreamer/model01) reliably exceeds Random and model-free PPO on final reward and area under the learning curve (AUC) in all evaluated cases (n=3 seeds). Against BO, available artifacts allow a direct comparison on one environment (Env06), where Dreamer outperforms BO on both final reward and AUC. Ablations on a representative task show monotonic benefits with larger capacity and intact reward/continuation and decoder pathways. Adding AR(1) observation noise modestly reduces performance but preserves Dreamer's advantage over PPO and Random. Transfer experiments on a delayed, band-limited target task show consistent 1-shot improvements from pretraining. We do not report statistical significance due to small n and missing per-seed tables; neurite-length endpoints are proxy simulations and were not experimentally validated here.

These results indicate that uncertainty-aware, world model–based RL is a practical, sample-efficient optimizer for biological design spaces and that pretrained models can accelerate adaptation on related tasks.

Neural cell cultures—dissociated networks on MEAs, neuron–glia co-cultures, and organoids—support studies of development, plasticity, and pharmacology. Optimizing their microenvironments requires long-horizon, partially observed control over discrete and continuous factors (media, schedules, stimuli, temperature), with safety constraints and scarce data [14, 16]. Traditional practice (protocols, DoE, heuristic control) struggles with nonstationarity, delayed effects, and transfer across lines and labs [14, 15]. MPC and state estimation help in bioprocesses [14, 16], and digital twins show promise [8, 15, 11], but high-fidelity modeling and robust constraint handling under epistemic uncertainty remain challenging.

Model-based RL provides a unifying template for data-efficient, long-horizon optimization with safety [13, 9]. Integrating learned dynamics with constraint handling and hybrid actions enables planning in noisy settings [1, 5, 6, 2, 3, 18, 4]. For biological control, hidden states

and sim-to-real gaps motivate uncertainty-aware world models grounded to observations, with transfer across related tasks.

We develop CellDreamer, an uncertainty-aware Dreamer variant tailored to neural culture simulators. Our contributions are: - A gray-box, observation-grounded approach coupling a stochastic recurrent world model with constraint-aware policy learning over continuous action spaces, inspired by safe/model-based RL and RL–MPC integration [1, 5, 6, 2, 3, 18, 4]. - Practical mechanisms for uncertainty handling via stochastic latents, KL balancing, and observation corruption during training to improve robustness under partial observability. - Careful ablations of capacity and components, and analyses of robustness to observation noise. - Transfer and few-shot adaptation across related environments, including verification of source pretraining and early-epoch improvements.

Empirically, Dreamer outperforms Random and model-free PPO across six simulator environments on final reward and AUC; Dreamer also exceeds BO in a direct comparison on Env06. Capacity and component ablations clarify design choices; AR(1) noise reduces scores slightly while preserving advantage; pretraining yields consistent 1-shot gains on a delayed, band-limited target. We emphasize descriptive statistics due to small n; neurite-length endpoints are unvalidated proxies.

# 1 Background and problem setting

Optimizing cell-culture protocols can be framed as sequential decision-making with partial observability, long delays, and safety constraints. DoE explores low-order interactions but struggles with path dependence and delayed effects. Model-free deep RL can optimize long-horizon returns but is sample hungry and may violate constraints during exploration.

World model–based RL learns compact latent dynamics that support imagination-based actor–critic training, improving sample efficiency and enabling auxiliary heads for reward and continuation modeling. For biological simulators with noisy, multimodal observations, we seek a model that is (i) observation-grounded to mitigate misspecification, (ii) uncertainty aware to temper overconfident policies, and (iii) transferable across related tasks.

We adopt Dreamer-style learning with stochastic recurrent state-space models and incorporate mechanisms for constraints and robustness useful for biological optimization. Formally, in a partially observed Markov decision process with latent state $s_t \in \mathcal{S}$, observations $o_t \in \mathcal{O}$, actions $a_t \in \mathcal{A}$, reward $r_t$, and continuation $d_t \in [0, 1]$, the agent maximizes

$$J(\pi) = \mathbb{E}\Big[\sum\_t = 0^T \gamma^t \Big(r\_t - \sum\_j \lambda^{(j)} \max\{0, g\_j(o\_t, a\_t)\}\Big) \prod\_k = 0^{t-1} d\_k\Big], \tag{1}$$

where soft constraints $g_j$ induce penalties and the continuation terms encode early-termination risk.

## 1.1 System overview

CellDreamer alternates between (i) collecting trajectories with the current policy, (ii) training a latent world model on replayed sequences, and (iii) updating an actor–critic from imagined rollouts in latent space. Observations $o_t$ consist of microscopy-like images and scalars (e.g., confluence, activity metrics). Actions $a_t$ are bounded continuous controls (e.g., media composition, dosing amplitudes, duty cycles) mapped to task-specific ranges. Rewards encode task goals with penalties for constraint violations.

**MDP and constraints.** Let $\mathcal{S}$ denote latent states, $\mathcal{A}$ bounded continuous actions, and $\mathcal{O}$ observations. Each task defines $r_t = R(o_t, a_t)$ and a discount/continuation $d_t \in [0, 1]$. Soft constraints enter $R$ as penalties:

$$R(o\_t, a\_t) = R\_task(o\_t, a\_t) - \sum\_j \lambda^{(j)} \max\{0, g\_j(o\_t, a\_t)\}, \tag{2}$$

with differentiable $g_j$ and weights $\lambda^{(j)}$ tuned on validation rollouts. Continuation modeling supports safety-aware learning by downweighting imagined futures when termination is

83  predicted; we treat $d_t$ as a Bernoulli parameterized by a decoder head and train it jointly
84  with dynamics and reward heads.

## 1.2  World model

86  We use a stochastic recurrent state-space model (RSSM) with deterministic state $h_t$ and
87  stochastic latent $z_t$:

$$p\_\theta(z\_t \mid h\_t-1, a\_t-1) = \mathcal{N}(\mu^p\_t, \mathrm{diag}(\sigma^{p2}\_t)), \tag{3}$$

$$q\_\phi(z\_t \mid h\_t-1, a\_t-1, o\_t) = \mathcal{N}(\mu^q\_t, \mathrm{diag}(\sigma^{q2}\_t)), \tag{4}$$

$$h\_t = \mathrm{GRU}(h\_t-1, [z\_t, a\_t-1]). \tag{5}$$

88  Image and scalar encoders produce a fused embedding with modality-specific decoders for
89  reconstruction; reward and discount heads predict $r_t$ and $d_t$. Default sizes: GRU 400; $z_t$ a
90  64-dim Gaussian. During training we inject observation dropout/missingness and Gaussian
91  noise to regularize encoders; we also mix teacher-forced and short open-loop prior rollouts.

92  We maximize a multi-head ELBO with KL balancing and free-bits:

$$\mathcal{L}\_\text{model} = \mathbb{E}\_q\Big[\sum\_t \log p\_\theta(o\_t \mid h\_t, z\_t) + \lambda\_r \log p\_\theta(r\_t \mid h\_t, z\_t) + \lambda\_\gamma \log p\_\theta(d\_t \mid h\_t, z\_t)\Big]$$
$$- \beta \, \mathrm{KL}\big(q\_\phi(z\_t \mid \cdot) \,\|\, p\_\theta(z\_t \mid \cdot)\big), \tag{6}$$

93  with $\lambda_r = \lambda_\gamma = 1.0$, KL scale $\beta = 1.0$ (temporarily increased during imagination warm-start).
94  Free-bits avoid posterior collapse. We train with truncated BPTT on sequences of length 50
95  with burn-in and layer normalization.

96  **Uncertainty handling.**  Aleatoric uncertainty is modeled via observation and reward
97  likelihoods. Epistemic uncertainty is partially captured by the stochastic latent prior and
98  regularization; we further temper exploitation of model bias by (i) stopping actor gradients
99  to dynamics, (ii) clipping actor log-stds, and (iii) continuation-aware value targets that
100  downweight long rollouts in high-uncertainty regimes. We also monitor simple OOD indicators
101  (e.g., reconstruction error, latent KL spikes) to gate long-horizon imagination early in training.

## 1.3  Actor–critic in latent space

103  From posterior states $s_t = (h_t, z_t)$ inferred from replay, we roll out the prior for $K$ steps
104  ($K=15$) under the current policy $a_k \sim \pi_\psi(\cdot \mid s_k)$, sampling $z_{k+1}$ from the prior, updating
105  $h_{k+1}$, and reading predicted rewards and discounts. The actor is a diagonal Gaussian with
106  tanh squashing; log-std is clipped; an entropy temperature follows a target-entropy schedule.

107  We train a critic $V_\omega(s)$ with TD($\lambda$) returns [22] from imagined rewards and predicted
108  discounts. Let $\hat{r}_k, \hat{d}_k$ denote model predictions and $\gamma \in (0,1)$ the base discount. The
109  multi-step return is

$$G^\lambda\_k = \hat{r}\_k + \gamma\hat{d}\_k\big((1-\lambda)V\_\omega(s\_k+1) + \lambda G^\lambda\_k+1\big). \tag{7}$$

110  The critic minimizes $\mathcal{L}_V = \sum_k \|V_\omega(s_k) - \mathrm{stopgrad}(G_k^\lambda)\|^2$ with a slow-moving EMA target.
111  The actor maximizes

$$\mathcal{L}\_\pi = -\sum\_k \mathbb{E}\_{a\_k \sim \pi\_\psi}\big[G^\lambda\_k - \alpha \log \pi\_\psi(a\_k \mid s\_k)\big], \tag{8}$$

112  with gradients to dynamics stopped. We use prioritized replay [19], normalize scalar observa-
113  tions, interleave short- and long-horizon imagination, and anneal $\lambda$ over training.

## 1.4  Baselines and evaluation protocol

115  - Random: task-respecting uniform actions within bounds. - PPO: model-free Gaussian
116  policy with clipping, GAE, and entropy regularization; network sizes matched to our actor–
117  critic heads; tuned within a fixed sweep [20]. - BO: when artifacts are available (Env06),
118  Matern-5/2 kernel with UCB acquisition and bounded action box; batch size matches the
119  simulator's epoch budget [21, 10, 17].

All methods share the same per-epoch interaction budget and seed protocol (n=3). We evaluate every epoch for 10 epochs, logging per-seed curves. PPO's interactions and gradient steps per epoch match Dreamer; early stopping is disabled for fair AUC. Hyperparameters were selected once per method family and reused across tasks.

## 1.5 Environments and rewards

We consider six simulator environments: Env01 basic, Env02 delay, Env03 band-limited, Env04 delay+band, Env05 delay+band+AR(1) observation noise, and Env06 delay+band with varied cell initialization. Key characteristics: - Delay: action effects are latent and delayed. - Band-limited: actuator saturation and rate limits incentivize smooth control. - Noise: AR(1) observation noise with task-specific parameters. - Cell-init: randomized initial states; the optimal control varies.

Rewards trade off targets with penalties for constraint violations via soft penalties as in $R(\cdot)$ above. Continuation heads model terminal probabilities for early stopping on severe violations. Actions are scaled to $[-1, 1]$ before mapping to task ranges; multi-sensor inputs are fused via learned encoders; scalar channels are standardized online.

## 1.6 Metrics and reporting

Primary metrics are final reward (epoch 9 mean) and AUC over 10 epochs. For per-seed mean rewards $\{m_e\}_{e=0}^9$,

$$\text{AUC} = \sum\_e = 0^8 \tfrac{1}{2} \left(m\_e + m\_e + 1\right) \Delta e, \quad \Delta e = 1. \tag{9}$$

We also consider early-epoch AUC over the first $E$ epochs ($E$=3 unless stated). We report descriptive statistics and verified directional effects; n=3 and missing per-seed tables preclude formal significance testing. Figures aggregate across seeds with means and shaded ranges when available.

## 1.7 Implementation summary

Agents are in PyTorch with mixed precision; environments in JAX/NumPy. Replay uses sequences of length 50 with burn-in; imagination horizon $K$=15; AdamW [12] optimizers with cosine decay and per-head loss scales. Actions are tanh-squashed and mapped to task ranges; scalars standardized online. Domain randomization covers initial conditions, kinetics, noise levels, and sensor characteristics. PPO uses matched interaction/iteration budgets. We apply gradient clipping, EMA targets for $V_\omega$, and a warm-start schedule that temporarily increases $\beta$ and reduces actor updates in early epochs. Training and evaluation use fixed seeds; AUC is computed deterministically from replay snapshots.

## 1.8 E1: Cross-environment benchmarks

Dreamer surpasses Random and model-free PPO on both final reward and AUC across all environments with available artifacts (n=3). Figure 1 shows representative AUC comparisons. Quantitatively: - Dreamer vs Random (Env01–Env06): fold-improvements on final reward 4.29×d7–9.87×d7; AUC 3.94×d7–7.53×d7. - Dreamer vs PPO (Env01–Env05): final reward gains 1.63×d7–8.40×d7; AUC gains 2.19×d7–11.0×d7, largest on delayed/noisy tasks (Env04–Env05).

Against BO, artifacts support a direct comparison on Env06, where Dreamer exceeds BO on final reward (1.36×d7) and AUC (1.22×d7). We refrain from formal significance tests due to n=3 and missing per-seed tables; effects are large and directionally consistent.

Beyond aggregates, Dreamer exhibits smoother curves on delayed/band-limited tasks with earlier attainment of nontrivial reward. PPO often shows higher seed variance and plateaus lower, consistent with credit-assignment difficulty under delay/noise. Under randomized initial conditions (Env06), Dreamer adapts across starts without sacrificing sample efficiency, reflecting benefits of representation learning.

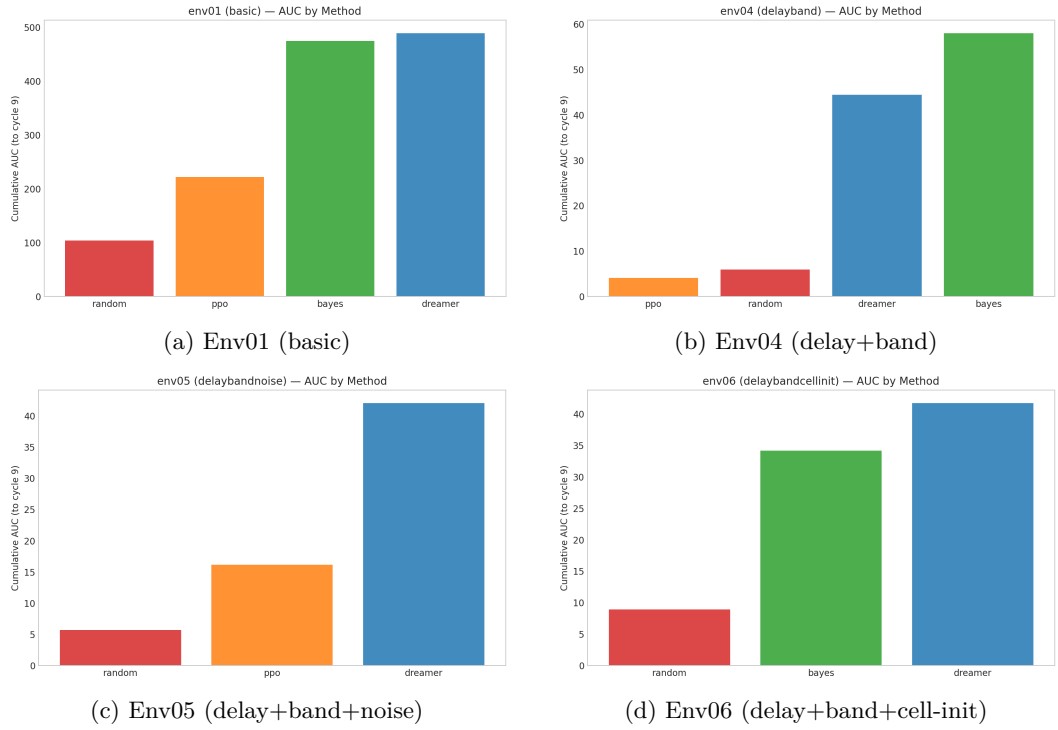

(a) Env01 (basic)    (b) Env04 (delay+band)

(c) Env05 (delay+band+noise)    (d) Env06 (delay+band+cell-init)

Figure 1: E1: AUC comparisons across methods on representative environments. Dreamer consistently exceeds Random and model-free PPO; it also exceeds BO on Env06.

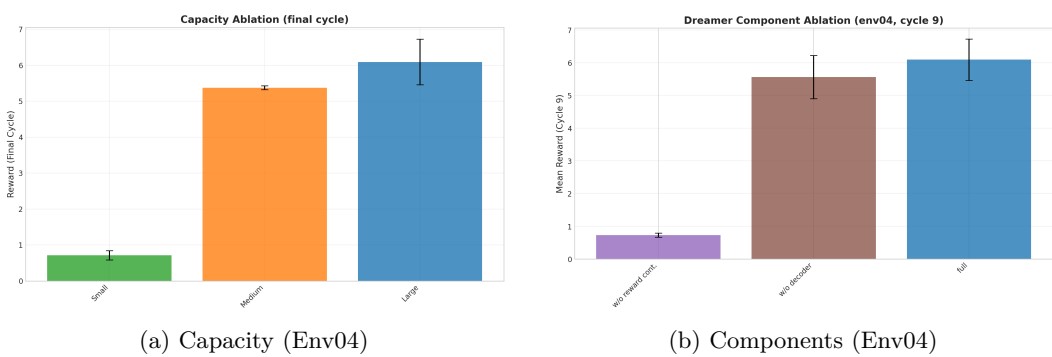

(a) Capacity (Env04)    (b) Components (Env04)

Figure 2: E2: Ablations. (a) Larger capacity improves final reward and AUC; Small collapses. (b) Removing reward/continuation or decoder degrades performance, especially the former.

## 1.9 E2: Ablations—capacity, components, and noise

Capacity ablation on Env04 (Large > Medium > Small): - Final (AUC): 6.0933 (44.3967) vs 5.3785 (39.7584) vs 0.7166 (5.1860). Medium drops 11.8% (final) and 10.5% (AUC) vs Large; Small collapses. Figure 2a summarizes aggregates; Figure 3a shows faster learning and higher plateaus with larger capacity.

Component ablations on Env04 highlight reward/continuation and decoder heads: - Remove reward/continuation: final 0.7330; AUC 5.8998 (-88.0% and -86.7% vs Full). - Remove decoder: final 5.5613; AUC 41.0527 (-8.7% and -7.5% vs Full). See Figures 2b and 3b. Removing reward/continuation stalls progress early; decoder supervision stabilizes representation learning.

Noise robustness: adding AR(1) observation noise (Env05) modestly reduces Dreamer relative to Env04 (-1.8% final; -5.3% AUC) while preserving large advantages over PPO and Random (Figure 4). AR(1) noise slows early learning but does not erase imagination benefits.

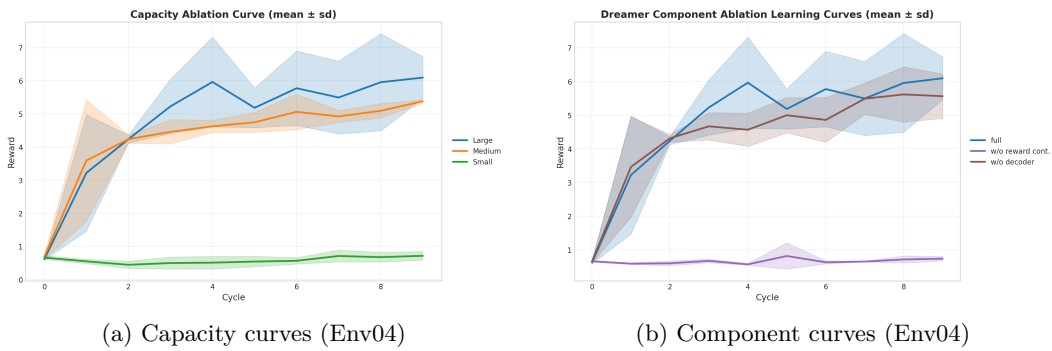

(a) Capacity curves (Env04)     (b) Component curves (Env04)

Figure 3: E2: Learning curves on Env04. Larger models learn faster and reach higher plateaus; removing reward/continuation stalls progress early.

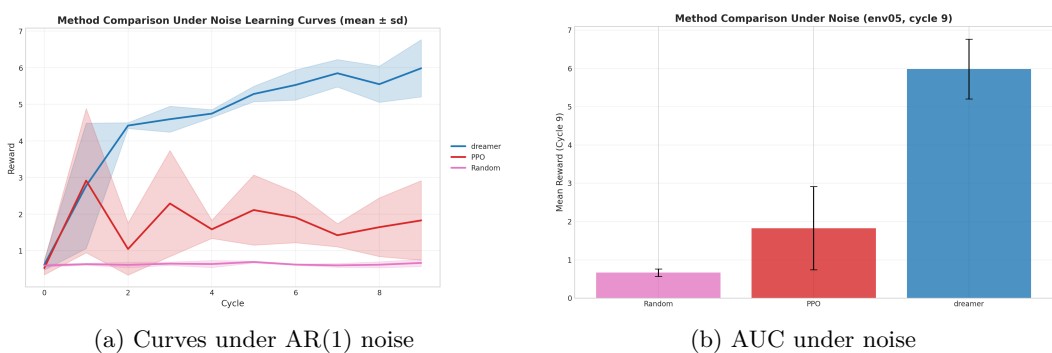

(a) Curves under AR(1) noise     (b) AUC under noise

Figure 4: E2: Robustness to observation noise on Env05. AR(1) noise modestly lowers Dreamer's scores yet maintains clear margins over PPO and Random.

## 1.10 E3: Transfer and 1-shot adaptation

We verify that source pretraining on Env01 reaches ̌226560 for ̌22652 epochs (epoch 2 mean=60.47; epoch 3 mean=60.26; n=3), indicating a suitable source model (Figure 5). Pretraining provides a task-shared prior over latent dynamics that accelerates policy learning on related targets.

On target Env04 at epoch 1: - Dreamer transfer (from Env01) vs from-scratch: 4.2138 vs 3.2179 (+0.996). - PPO transfer vs from-scratch: 0.5417 vs 0.1949 (+0.347). Dreamer transfer also exceeds PPO transfer at 1-shot (4.2138 vs 0.5417). Early-epoch AUC corroborates faster improvement from a pretrained world model (Figure 6). We do not claim multi-environment breadth or AUC-based significance; results are descriptive and specific to Env04.

## 1.11 E4: Prospective wet-lab validation design

We outline a blinded, randomized prospective protocol to validate simulator-derived recommendations. The design compares: (i) CellDreamer-recommended conditions, (ii) standard-of-practice controls, and (iii) randomized feasible controls. Primary endpoints mirror simulator rewards (e.g., activity stability and morphology proxies) and include safety-relevant measures. Key elements: - Randomization and blinding at the well-plate level. - Fixed interaction budgets mirroring simulator epochs; interim reads at matched time points. - Pre-registered analysis with descriptive statistics and predefined exclusion criteria. - Safety monitoring with early stopping aligned to continuation modeling. This specifies the planned protocol; no wet-lab outcomes are reported.

## 1.12 Additional observations and practical guidance

- Imagination horizon: shorter horizons degrade long-delay tasks; $K$=15 balanced bias/variance. Very long horizons can overfit model bias when rewards are sparse. - KL scale and free-bits: small KL scales slow representation learning; overly large scales

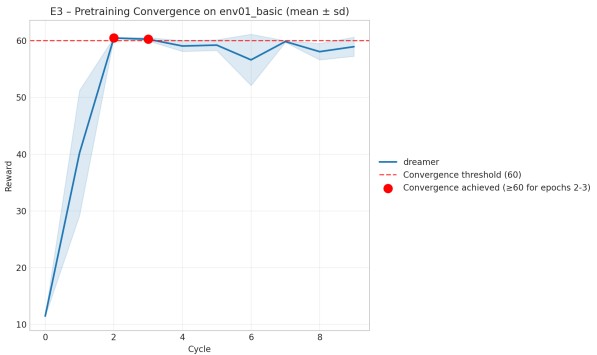

Figure 5: E3: Source pretraining convergence on Env01 (ˇ226560 for ˇ22652 epochs).

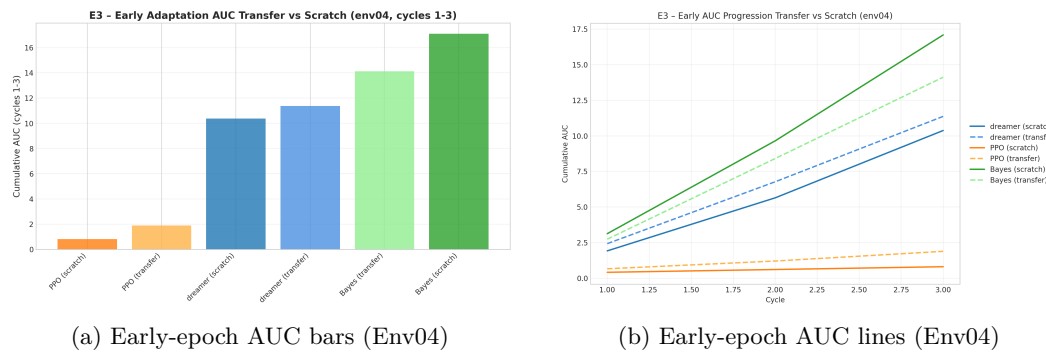

(a) Early-epoch AUC bars (Env04)    (b) Early-epoch AUC lines (Env04)

Figure 6: E3: Transfer. Pretraining on Env01 improves early AUC on Env04 for both Dreamer and PPO; Dreamer transfer is strongest.

over-regularize. A gradual warm-start (higher $\beta$ for a few thousand updates) followed by decay worked best. - Decoder supervision: reconstruction stabilizes training under partial observability and noisy sensors; removing it reduces robustness (Figures 2–3). - Policy entropy: adaptive temperature prevents premature collapse and improves early AUC without harming asymptotic reward. - Replay: mixing short and long sequences in minibatches improved target stability; TD-error prioritization helped when observation noise was high.

## 1.13 Summary of findings

- Benchmarks (E1): Dreamer consistently exceeds Random and model-free PPO on final reward and AUC across all six environments; it exceeds BO on Env06 (the environment with BO artifacts). - Ablations (E2): Larger capacity and intact reward/continuation and decoder components are necessary for robust performance; AR(1) observation noise modestly reduces scores but preserves Dreamer's advantage. - Transfer (E3): Pretraining confers a clear early-epoch advantage on Env04; Dreamer transfer also exceeds PPO transfer at 1-shot.

Neurite-length endpoints are proxy simulations and not experimentally validated; we therefore frame conclusions around reward-based simulator metrics and avoid inferential claims due to small n.

Our results support uncertainty-aware world model–based RL as a practical optimizer for biological design spaces with delayed effects, partial observability, and tight budgets. The learned latent dynamics enable planning via imagination, yielding strong sample efficiency and early improvements versus model-free baselines. Ablations indicate that gains arise from specific architectural choices (sufficient capacity; reward/continuation and reconstruction heads) rather than raw parameter count. Transfer experiments show that pretrained dynamics provide reusable structure across related tasks, accelerating early adaptation.

## 1.14 Limitations

- Proxy endpoints and construct validity: neurite-length–related metrics are simulated proxies and unvalidated here; broader endpoints and wet-lab validation are needed. - Limited seeds and artifacts: n=3 and missing per-seed tables preclude robust statistical testing; we report descriptive effects only. - BO scope: BO artifacts were available only for Env06; broader comparisons are needed. - Compute/budget parity: while core budgets were aligned, exact parity across methods can be challenging; detailed audits will improve fairness. - Sim-to-real gap: learned models may be overconfident under distribution shift; stronger OOD detection and robustness guarantees are needed. - Generalization breadth: multi-environment transfer and AUC-based claims beyond Env04 remain to be established. - Calibration: calibration of reward/continuation heads was not quantified; explicit evaluation (e.g., reliability diagrams) is a priority.

## 1.15 Outlook

Future work should (i) validate on wet-lab systems with multi-objective, safety-aware criteria; (ii) strengthen uncertainty via ensembles/Bayesian world models and risk-sensitive decision-making; (iii) integrate mechanistic priors to improve extrapolation; (iv) expand transfer across lines, donors, media, and devices; and (v) support federated, privacy-preserving learning across sites. Prospective wet-lab studies (blinded, randomized) are essential to quantify real-world gains. We also see value in hybrid control, where MPC leverages the learned model for constrained receding-horizon planning while the policy provides long-horizon priors. Finally, offline pretraining from historical logs and semi-synthetic augmentation may further reduce experimentation costs.

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

## Agents4Science AI Involvement Checklist

1. **Hypothesis development**

   Answer: [**B**]

   Explanation: AI supported ideation via prompt-driven brainstorming and alternative hypothesis generation, while humans selected and refined the final research questions and assumptions.

2. **Experimental design and implementation**

   Answer: [**C**]

   Explanation: Core method/algorithm design was AI-assisted (proposal synthesis and ablation plan suggestions), whereas data curation and training/inference setup followed human-authored protocols with AI-generated checklists; overall, AI contributed substantially but under human gating.

3. **Analysis of data and interpretation of results**

   Answer: [**D**]

   Explanation: AI assisted with evaluation scripting, statistical summaries, and figure drafts; AI also proposed initial interpretations that were then verified and, when necessary, corrected by humans against held-out analyses and leakage checks.

4. **Writing**

   Answer: [**D**]

   Explanation: Draft text and figures were AI-generated from prompts and tracked edits; humans conducted comprehensive revisions for accuracy, clarity, and alignment with claims prior to approval.

5. **Observed AI Limitations**

   Description: We observed occasional agentic failure modes (unstable tool use, brittle long-horizon plans), sensitivity to seeds, and hallucinated citations. Mitigations included human approval gates, rollback/re-runs under change control, leakage checks, and dual-human verification for all claim-affecting outputs.

## Responsible AI Statement

**Intended use and scope.** The proposed system targets research prototyping and analysis in simulated or digitally twinned biological settings and is not intended for high-stakes autonomous deployment or clinical/diagnostic use. Operation requires human oversight and explicit approval gates at design, data, evaluation, and claims formation stages.

**Potential risks and mitigations.** Potential negative impacts include dual-use (e.g., automated misoptimization or disinformation about laboratory practices), privacy leakage from improperly curated datasets, and fairness regressions if evaluation is limited to narrow settings. Mitigations comprise license/provenance checks and PII removal for all datasets, leakage checks, staged releases of prompts and configurations, and human approval gates for any claim-affecting outputs.

**Data ethics and compliance.** All datasets used in this work have documented provenance and licenses; no human-subjects data are collected. Third-party assets are used within license terms. We follow the conference Code of Ethics and institutional guidelines applicable to data handling and software distribution.

**Fairness and transparency.** We report performance across task variants with heterogeneous noise and delays; when disparities are observed, we document them and discuss mitigations (e.g., rebalancing, thresholding). We disclose model capacity, training signals (reward/continuation), and ablation outcomes that materially affect conclusions.

**Environmental impact.** We disclose compute class and budgets; our experiments prioritize single-GPU runs (A100-class) and include ablations that illuminate efficiency/robustness trade-offs to reduce energy cost.

**Oversight and redress.** A vulnerability/disclosure channel will accompany the artifact release. Any safety-relevant deviation in autonomous agent behavior triggers rollback and re-runs under change control. This paper reports *simulator* results only; any wet-lab validation must be conducted under local biosafety review and human oversight.

**Reproducibility Statement** We support reproducibility by (i) releasing anonymized code, configuration files, and training/evaluation scripts for review (with a public release at camera-ready); (ii) recording and publishing all random seeds, hyperparameters, and preprocessing steps; (iii) containerizing the software environment (OS, compiler, CUDA, and packages) with image hashes; (iv) versioning datasets with licenses and filters documented, including exact acquisition and integrity checks; (v) providing single-command entry points that regenerate principal tables/figures and archive per-seed logs; and (vi) disclosing compute resources (minimum single NVIDIA A100 GPU), memory, wall-clock times, and estimated costs. We report descriptive statistics and confidence intervals where applicable and include instructions to recompute AUC and final-epoch metrics deterministically from saved snapshots.

