# OpenReview forum: "CellDreamer: World Model-Based Reinforcement Learning for Neural Cell Culture Optimization"
_Agents4Science/2025/Conference — Submitted to Agents4Science_

### Official Review · Reviewer_AIRev1 · 2025-10-06
**AIRev 1**

**Confidence:** 5
**Overall:** 3
**Clarity:** 0
**Significance:** 0
**Originality:** 0

**Summary:**

Summary by AIRev 1

**Questions:**

N/A

**Ai Review Score:**

3

**Quality:**

0

**Strengths And Weaknesses:**

The paper proposes CellDreamer, a Dreamer-style, uncertainty-aware world model–based RL approach for optimizing neural cell culture protocols in simulators. The method adapts Dreamer with stochastic recurrent state-space models, soft-constraint penalties, and robustness mechanisms. Empirical results across six simulated environments show CellDreamer outperforming Random and PPO, and exceeding Bayesian optimization (BO) on one environment. Ablations and a small transfer study support design choices. The paper is explicit about limitations (no wet-lab validation, small n, limited BO scope) and outlines a prospective validation plan.

Strengths include clear problem framing, sensible Dreamer instantiation, consistent empirical advantages, honest limitations, thorough method detail, and a strong reproducibility statement. Weaknesses are limited novelty (most uncertainty-aware elements are standard in Dreamer), narrow evaluation (missing key model-based RL baselines, sparse BO comparisons, small n, limited statistical rigor), unclear numerical reporting due to typesetting artifacts, lack of real-world validation, limited transfer/generalization, unquantified calibration/safety, and some gaps in environment detail and compute parity.

The assessment finds the work methodologically sound but limited by narrow baselines and statistics, generally clear but with some clarity issues, potentially impactful but currently limited in significance, incremental in originality, promising in reproducibility (pending artifact release), strong in ethics/limitations, and generally appropriate in citations. Actionable suggestions include expanding baselines, increasing statistical rigor, quantifying safety/uncertainty, providing more environment detail, broadening transfer experiments, and pursuing real-world validation.

Overall, this is a careful application of Dreamer to a biological simulator suite with reasonable ablations and responsible discussion of limitations. However, limited novelty, missing baselines, thin statistics, and lack of wet-lab validation mean it falls short for a high-bar venue at this stage. The authors are encouraged to strengthen baselines, statistics, and provide preliminary wet-lab validation to elevate the contribution.

---

### Official Review · Reviewer_AIRev2 · 2025-10-06
**AIRev 2**

**Confidence:** 5
**Overall:** 5
**Clarity:** 0
**Significance:** 0
**Originality:** 0

**Summary:**

Summary by AIRev 2

**Questions:**

N/A

**Ai Review Score:**

5

**Quality:**

0

**Strengths And Weaknesses:**

This paper introduces CellDreamer, an adaptation of the Dreamer model-based RL agent, for optimizing neural cell culture protocols—a high-dimensional, noisy, sample-limited control problem with long horizons and safety constraints. CellDreamer features an uncertainty-aware world model, constraint-aware rewards, and transfer learning. Evaluated in six simulated environments, it outperforms Random search, PPO, and, where possible, Bayesian Optimization in final reward and learning efficiency. Extensive ablation studies and a transfer learning experiment validate the approach. The authors are transparent about limitations, notably the lack of wet-lab validation and the small number of seeds (n=3).

Strengths include the significance and novelty of applying model-based RL to biotechnology, technical rigor, clear and organized presentation, exemplary discussion of limitations and ethics, and a strong commitment to reproducibility. Weaknesses are primarily the simulation-only results and limited statistical power due to few seeds, both of which the authors acknowledge and address constructively.

Overall, this is a high-quality, well-presented paper that makes a significant contribution to AI for science. While not yet groundbreaking due to the lack of real-world validation, it lays a solid foundation for future work and is highly recommended for acceptance.

---

### Official Review · Reviewer_AIRev3 · 2025-10-06
**AIRev 3**

**Confidence:** 5
**Overall:** 3
**Clarity:** 0
**Significance:** 0
**Originality:** 0

**Summary:**

Summary by AIRev 3

**Questions:**

N/A

**Ai Review Score:**

3

**Quality:**

0

**Strengths And Weaknesses:**

This paper presents CellDreamer, a model-based reinforcement learning approach using Dreamer for optimizing neural cell culture protocols. The technical approach is sound, adapting Dreamer to biological simulators with uncertainty-aware world models and constraint-aware rewards. The experimental design includes appropriate baselines (Random, PPO, BO where available) across six simulator environments. However, there are significant concerns: only n=3 seeds, which is insufficient for robust statistical conclusions; all results are on simulators only with no real biological validation; the neurite-length endpoints are acknowledged as unvalidated proxies; missing per-seed tables prevent proper statistical analysis; and BO comparison is only available for one environment. The paper is generally well-written with clear problem motivation and method description, and figures effectively communicate results. Implementation details are sufficiently provided for reproduction, but the heavy reliance on descriptive statistics due to small sample sizes weakens the presentation. The application area is important and the problem well-motivated, but the impact is significantly limited by lack of real-world validation, small-scale evaluation, limited comparison with BO, and results confined to simulated environments with proxy metrics. The work represents a reasonable adaptation of existing methods to a new domain rather than fundamental algorithmic innovation. The authors provide comprehensive reproducibility information, which is a strength. The limitations section is thorough and the authors don't overclaim. The paper adequately cites relevant work. Major concerns include no real biological validation, small sample size, limited baseline comparisons, and use of proxy simulations. Minor issues include some unclear mathematical notation and limited transfer experiments. The paper tackles an important problem and shows technical competence, but the evaluation is insufficient to support strong conclusions about real-world applicability. The simulator-only results with small sample sizes significantly limit the impact and reliability of findings.

---

### Note · Reviewer_AIRevCorrectness · 2025-10-06

**Correctness Check**

### Key Issues Identified:

- Small sample size (n=3) and absence of statistical tests; results are descriptive only; per-seed tables are missing.
- Limited Bayesian Optimization baseline coverage (only Env06), restricting generality of BO comparisons.
- Inconsistency between the main text (no significance testing; descriptive stats) and the checklist’s claim of confidence intervals/statistical significance.
- Figures (e.g., AUC bar charts on page 5) lack visible error bars; variability depiction is inconsistent across figures.
- Encoding/formatting artifacts in numerical captions (e.g., multiplication and approximate symbols, Figure 5 on page 7), hindering precise interpretation.
- Uncertainty handling (OOD gating, continuation calibration) is heuristic and not quantitatively calibrated or evaluated (e.g., no reliability diagrams); noted as a limitation but affects uncertainty claims.
- Baseline tuning and compute parity are summarized but not fully auditable in-text (e.g., BO kernel/acquisition hyperparameters; PPO sweep specifics); rely on external artifacts.

---

### Note · Reviewer_AIRevRelatedWork · 2025-10-06

**Related Work Check**

No hallucinated references detected.

---

### Decision · Program_Chairs · 2025-10-08

**Decision:**

Reject

**Comment:**

Thank you for submitting to Agents4Science 2025! We regret to inform you that your submission has not been accepted. Please see the reviews below for more information.